# OVERCONFIDENCE IN LLM-AS-A-JUDGE: DIAGNOSIS AND CONFIDENCE-DRIVEN SOLUTION

## ABSTRACT

Large Language Models (LLMs) are widely used as automated judges, where practical value depends on both accuracy and trustworthy, risk-aware judgments. Existing approaches predominantly focus on accuracy, overlooking the necessity of well-calibrated confidence, which is vital for adaptive and reliable evaluation pipelines. In this work, we advocate a shift from accuracy-centric evaluation to confidence-driven, risk-aware LLM-as-a-Judge systems, emphasizing the necessity of well-calibrated confidence for trustworthy and adaptive evaluation. We systematically identify the **Overconfidence Phenomenon** in current LLM-as-a-Judges, where predicted confidence significantly overstates actual correctness, undermining reliability in practical deployment. To quantify this phenomenon, we introduce **TH-Score**, a novel metric measuring confidence-accuracy alignment. Furthermore, we propose **LLM-as-a-Fuser**, an ensemble framework that transforms LLMs into reliable, risk-aware evaluators. Extensive experiments demonstrate that our approach substantially improves calibration and enables adaptive, confidence-driven evaluation pipelines, achieving superior reliability and accuracy compared to existing baselines.

## 1 INTRODUCTION

The widespread adoption of large language models (LLMs) as automated judges—termed the LLM-as-a-Judge paradigm—has revolutionized the evaluation of AI-generated content by offering scalability and efficiency over traditional human annotation (Zheng et al., 2023). In this paradigm, LLMs act as evaluators, with one common application being pairwise comparisons where the model decides which of two text segments is better based on criteria like quality, relevance, or coherence. However, the practical value of these systems depends not only on accuracy but also on trustworthy, risk-aware judgments that can adapt to real-world deployment scenarios. Existing approaches, such as FairEval (Wang et al., 2023a) and JudgeBench (Tan et al., 2024), predominantly emphasize accuracy, often overlooking the critical role of well-calibrated confidence. This calibration, defined as the alignment between a model's predicted confidence and its actual correctness, is essential for building adaptive evaluation pipelines. For instance, well-calibrated confidence allows high-confidence outputs to be automatically accepted, minimizing manual intervention, while low-confidence cases can be flagged for human review (Li et al., 2024). In this work, we advocate a fundamental shift from accuracy-centric evaluations to confidence-driven, risk-aware LLM-as-a-Judge systems, prioritizing calibration to ensure reliable and trustworthy assessments.

Despite these potential benefits, current LLM-as-a-Judge systems suffer from a pervasive Overconfidence Phenomenon, where predicted confidence levels significantly overstate actual correctness (Mielke et al., 2022; Zhou et al., 2023), thereby undermining reliability in practical applications. Through systematic analysis, we observe that state-of-the-art LLMs exhibit this issue prominently, leading to inflated confidence scores that do not reflect true performance (Zhao et al., 2021). This misalignment results in substantial risks: overconfident models may propagate erroneous judgments without detection, eroding the efficiency gains of automated evaluation, while also complicating downstream decision-making in pipelines (Gu et al., 2024). Furthermore, existing benchmarks and metrics exacerbate the problem by focusing on aggregate accuracy without addressing confidence alignment, introducing biases such as response length or model familiarity that distort calibration assessments (Chen et al., 2024; Zheng et al., 2023; Wang et al., 2023a). Consequently, the lack

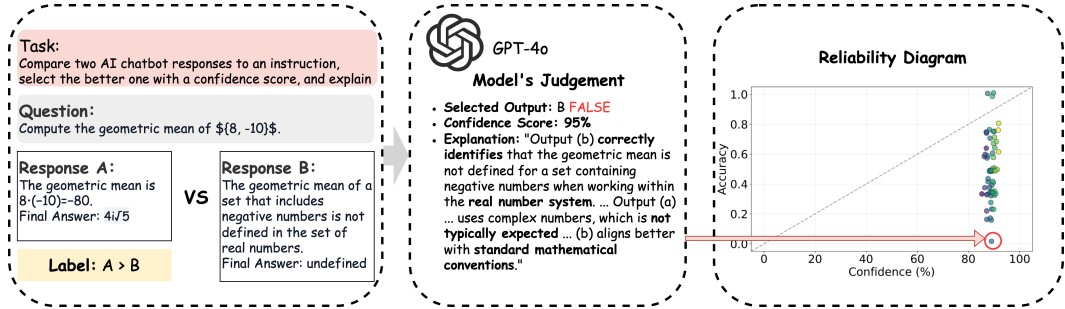

Figure 1: Overconfidence phenomenon of GPT-4o as a Judge on JudgeBench, illustrated with a specific example (content omitted for demonstration purposes) and a Reliability Diagram. The Reliability Diagram plots the model's confidence scores against actual accuracy, revealing calibration gaps where overconfidence occurs.

of calibration-aware tools limits the deployment of LLMs as dependable evaluators in high-stakes environments.

To address these challenges, we introduce TH-Score, a novel metric that quantifies confidence-accuracy alignment by focusing on critical high- and low-confidence intervals, where practical decisions hinge. Unlike traditional metrics like accuracy or Expected Calibration Error (ECE)—which ignore confidence or overlook key thresholds—TH-Score balances accuracy within these intervals against their coverage, rewarding aligned successes (e.g., high-confidence correct predictions) while penalizing mismatches like overconfident errors. This makes TH-Score a principled tool for detecting the Overconfidence Phenomenon under LLM-as-a-Judge scenario, highlighting cases where high confidence fails to match actual correctness.

Furthermore, we propose LLM-as-a-Fuser, an ensemble framework that leverages a dedicated "fuser" LLM to synthesize judgments and critiques from multiple models, transforming LLMs into reliable, risk-aware evaluators. By integrating diverse perspectives, LLM-as-a-Fuser significantly enhances calibration. Extensive experiments on a widely-used benchmark demonstrate that our approach achieves superior calibration, reliability, and overall accuracy compared to existing baselines, paving the way for more trustworthy LLM-as-a-Judge systems in practical settings.

In a nutshell, our contributions are threefold:

- **Overconfidence Phenomenon**: We identify and characterize the overconfidence in LLM-as-a-Judge, where confidence overstates correctness, limiting risk-aware evaluation.
- **Metric Innovation**: We introduce TH-Score, a novel metric quantifying confidence-accuracy alignment over specified intervals for trustworthy LLM-as-a-Judge.
- **Framework Advancement**: We propose LLM-as-a-Fuser, an ensemble method boosting calibration for adaptive, confidence-driven pipelines with higher reliability and accuracy.

## 2 RELATED WORK

### 2.1 LLM-AS-A-JUDGE

LLMs are increasingly used as automated evaluators for text quality. Zheng et al. (2023) showed GPT-4 aligns with human judgments in pairwise comparisons, but proprietary APIs limit reproducibility. PandaLM (Wang et al., 2023b) introduced a 7B-parameter local evaluator with 94% agreement with ChatGPT, supporting offline use. JudgeLM (Zhu et al., 2025) and Agent-as-a-Judge (Zhuge et al., 2024) use modular frameworks with memory and planning, cutting DevAI evaluation costs by 97%. However, alignment between model confidence and accuracy is often ignored, causing inconsistent judgments. Meta's self-rewarding models (Wu et al., 2024) generate and evaluate outputs iteratively, but calibration needs further study.

As LLM-based judges gain traction for evaluating and enhancing LLMs, various benchmarks have emerged to gauge their effectiveness. Prior works like LLMEval (Lin & Chen, 2023), MTBench,

and FairEval primarily assess how well LLM-based judges align with subjective human preferences, often emphasizing stylistic differences over factual and logical accuracy. Similarly, LLMBar (Zeng et al., 2024) evaluates judges based on their ability to follow instructions, using response pairs with explicit ground truth labels tied to instruction adherence. In contrast, JudgeBench offers a novel benchmark specifically designed to test LLM-based judges' reasoning capabilities. It features 350 carefully curated challenging response pairs across knowledge, reasoning, math, and coding domains, each containing one objectively correct and one subtly incorrect response, prioritizing factual and logical correctness over subjective or stylistic factors.

## 2.2 CALIBRATION IN LLMS

Accurate calibration, aligning a model's confidence with its accuracy, is crucial for reliable LLM applications. Traditional methods like temperature scaling (Guo et al., 2017) adjust confidence with a single scalar but are less effective for large models, while Bayesian methods are computationally infeasible. Recent approaches, such as the Thermometer method (Shen et al., 2024), train auxiliary models for recalibration, achieving top uncertainty quantification across 12 benchmarks, and SPACE (Yi et al., 2024) uses lightweight linear layers for dynamic confidence adjustment. However, most techniques focus on single models, missing multi-model aggregation benefits, and Collaborative Calibration (Yang et al., 2024) reduces overconfidence via multi-agent deliberation but requires significant resources. Current research lacks focus on calibration's impact on downstream tasks like data generation, where confidence filtering affects output quality, warranting further exploration.

## 2.3 UNCERTAINTY QUANTIFICATION AND REWARD MODELING

Uncertainty-aware frameworks bridge calibration and practical applications. Generating with Confidence (Lin et al., 2023) combines Monte Carlo dropout and response length analysis to filter low-confidence outputs, demonstrating that well-calibrated models yield higher-quality synthetic data. Inference-Time Scaling (Liu et al., 2025) dynamically aligns reward models with human preferences, indirectly improving calibration through gradient-free optimization. However, these approaches often assume static datasets, failing to address the iterative nature of LLM-as-a-Judge workflows. Benchmarking LLMs via Uncertainty Quantification (Ye et al., 2024) reveals that calibration degrades under distribution shifts, underscoring the need for adaptive methods.

## 3 OVERCONFIDENCE IN LLM-AS-A-JUDGE

In the LLM-as-a-Judge paradigm, models are typically required to select the superior option from pairwise samples. However, the reliability of model predictions warrants careful examination, particularly regarding the Overconfidence Phenomenon—a tendency for language models to display predicted confidence levels that significantly exceed their actual accuracy, resulting in calibration gaps that undermine reliability. Underconfident models tend to underestimate their own accuracy, while overconfident ones overestimate their judgment correctness. Such biases introduce noisy signals that can adversely affect the performance of downstream tasks (e.g., reward modeling). Particularly in unsupervised or weakly-supervised scenarios, developing a well-calibrated model where judgment capability aligns with confidence becomes crucial. By acquiring confidence of model judgments, we can not only filter out low-accuracy predictions but also effectively identify high-accuracy decisions, thereby enhancing the overall system reliability.

### 3.1 HOW TO MEASURE CONFIDENCE IN LLMS?

We employed three methods for calculating confidence: Self-Confidence (SC), Multiple-Prompting (MP) confidence, and Logprob (LogP) confidence.

**SC setting**: We prompt the model to output both the result and its confidence. Model's temperature is set to 0 to ensure the reproducibility of the setting.

**MP setting**: We adopt a method similar to SimpleQA (Wei et al., 2024), but reduce the number of requests from 100 to 10 for efficiency, while keeping the temperature at 0.7. The final reply is determined by majority voting, and the confidence is the count of the chosen response over 10.

**LogP Setting.** In this setting, confidence scores are derived from softmax-normalized logits for the final output tokens (e.g., 'A' or 'B'). For a binary choice task with options $A$ and $B$, and corresponding logits $l_A$ and $l_B$, we first compute the softmax probabilities:

$$p(A) = \frac{e^{l_A}}{e^{l_A} + e^{l_B}}, \quad p(B) = \frac{e^{l_B}}{e^{l_A} + e^{l_B}}.$$

Confidence is then defined as the maximum probability:

$$\text{Confidence}_{\text{logp}} = \max\left(p(A), p(B)\right).$$

Temperature is set to 0 to ensure deterministic outputs.

Figure 2: Visualization of the three confidence calculation settings: Self-Confidence (SC), Multiple-Prompting (MP), and Logprob (Logp), using data with ID 122 from JudgeBench as an example.

## 3.2 EXISTING CALIBRATION EVALUATION METRICS

To conduct an objective and comprehensive evaluation of the calibration of LLM-as-a-Judge, we applied five existing metrics—Expected Calibration Error (ECE), Adaptive Calibration Error (ACE), Maximum Calibration Error (MCE), Brier Score, and Negative Log Likelihood (NLL)—to the three confidence calculation methods described earlier in this section. Table 3 provides a brief introduction to the calculation methods and characteristics of these metrics.

## 3.3 INITIAL RESULTS

We systematically evaluate 14 cutting-edge models on the JudgeBench benchmark, with complete results presented in Table1 and 2. These include open-source models such as Qwen3-235B-A22B (Qwen Team, 2025), DeepSeek-R1-0528 (DeepSeek-AI et al., 2025), R1-Distill-Qwen, R1-Distill-Llama, DeepSeek-V3-0324 (DeepSeek-AI et al., 2024), Llama-3.3-70B (Dubey et al., 2024), and Mistral-Nemo (Team, 2024), as well as proprietary models like OpenAI-o3-mini (OpenAI, 2025b), Claude-Sonnet-4 (Anthropic, 2025), GPT-4.1 (OpenAI, 2025a), GPT-4.1-mini, Gemini-2.5-Flash (DeepMind, 2025), GPT-4o (Ahmad et al., 2024), and GPT-4.1-nano. Our analysis focuses on the impact of model scales on accuracy and confidence calibration (ECE/ACE), visually further illustrated by reliability plots in Figure3 showcasing calibration gaps in high-confidence (red) and low-confidence (green) regions for selected models.

## 3.4 EMPIRICAL OBSERVATIONS OF OVERCONFIDENCE

Figure 3 reveals significant calibration gaps across the evaluated models, with most exhibiting the Overconfidence Phenomenon in high-confidence regions (green). This pattern undermines the relia-

Table 1: Performance metrics of various models under the Self-Confidence (SC) setting, categorized into Open Source and Proprietary models. Arrows indicate optimization direction: ↑ higher is better, ↓ lower is better. Best results for each metric are bolded.

| Model | Acc ↑ | ECE ↓ | ACE ↓ | Brier Score ↓ | MCE ↓ | NLL ↓ | TH ↑ |
|---|---|---|---|---|---|---|---|
| **Open Source Models** | | | | | | | |
| Qwen3-235B-A22B | **77.43** | **11.78** | 12.16 | 0.16 | 63.50 | 0.52 | **17.52** |
| DeepSeek-R1-0528 | 76.86 | 12.07 | **11.39** | **0.13** | **40.00** | **0.42** | 14.59 |
| R1-Distill-Qwen | 65.71 | 27.26 | 27.10 | 0.29 | 69.00 | 0.91 | 8.16 |
| R1-Distill-Llama | 59.71 | 31.02 | 30.89 | 0.31 | 65.00 | 1.31 | 7.01 |
| DeepSeek-V3-0324 | 49.71 | 36.21 | 36.35 | 0.37 | 50.24 | 1.03 | 2.46 |
| Llama-3.3-70B | 42.00 | 47.37 | 46.78 | 0.45 | 63.78 | 2.75 | 0.80 |
| Mistral-Nemo | 20.29 | 74.22 | 74.21 | 0.71 | 80.00 | 3.01 | -11.64 |
| **Proprietary Models** | | | | | | | |
| OpenAI-o3-mini | 74.29 | 15.97 | 17.20 | 0.20 | 60.00 | 0.62 | 12.83 |
| Claude-Sonnet-4 | 64.29 | 17.98 | 18.00 | 0.24 | 45.00 | 0.69 | 9.89 |
| GPT-4.1 | 63.14 | 26.39 | 26.86 | 0.29 | 55.00 | 0.85 | 7.55 |
| GPT-4.1-mini | 55.71 | 32.70 | 32.79 | 0.35 | 44.21 | 1.00 | 3.29 |
| Gemini-2.5-Flash | 39.43 | 30.49 | 30.41 | 0.26 | 56.11 | 0.78 | 2.71 |
| GPT-4o | 49.71 | 39.25 | 39.28 | 0.40 | 57.50 | 1.15 | 1.57 |
| GPT-4.1-nano | 26.86 | 57.03 | 57.08 | 0.52 | 72.50 | 1.38 | -0.07 |

(a) DeepSeek-R1-0528     (b) Mistral-Nemo     (c) DeepSeek-V3-0324

(d) Claude-Sonnet-4     (e) Gemini-2.5-Flash     (f) GPT-4o

(g) Qwen3-235B-A22B     (h) GPT-4.1     (i) Llama-3.3-70B

Figure 3: Illustration of calibration gaps in low-confidence regions (red) and high-confidence regions (green) where models show significant accuracy-confidence discrepancy.

bility of the LLM-as-a-Judge, as models like DeepSeek-R1-0528 and GPT-4o cluster predictions at high confidence levels (90-100%) but achieve accuracies well below the ideal calibration line.

This overconfidence impacts downstream tasks, such as data filtering, by retaining flawed outputs (false positives) or discarding valuable ones (false negatives), thereby degrading overall performance. For instance, high ECE values in GPT-4o (39.25 in SC, 47.09 in MP, 45.05 in LogP),

Mistral-Nemo (74.22 in SC, 68.89 in MP, 64.63 in LogP), and GPT-4.1-nano (57.03 in SC, 67.43 in MP, 66.05 in LogP) necessitate increased human oversight to mitigate risks, diminishing the efficiency of automated judging processes (see Table 1 and Appendix A.5).

Table 2: Performance comparison of different LLMs under logP setting.

| Model | Acc ↑ | ECE ↓ | ACE ↓ | Brier Score ↓ | MCE ↓ | NLL ↓ | TH Score ↑ |
|---|---|---|---|---|---|---|---|
| DeepSeek-R1-0528 | **78.29** | **6.84** | **6.62** | **0.1298** | **46.15** | **0.4211** | 2.96 |
| GPT-4.1 | 63.43 | 34.46 | 34.43 | 0.3462 | 63.80 | 1.7287 | **7.36** |
| GPT-4.1-mini | 55.14 | 42.56 | 42.53 | 0.4253 | 58.32 | 1.7946 | 2.56 |
| GPT-4o | 50.86 | 45.05 | 45.06 | 0.4493 | 61.19 | 1.6238 | 0.79 |
| DeepSeek-V3-0324 | 48.29 | 50.76 | 50.68 | 0.5044 | 50.76 | 2.4714 | -0.85 |
| Llama-3.3-70B | 43.43 | 53.53 | 53.55 | 0.5318 | 54.04 | 2.3400 | -3.25 |
| GPT-4.1-nano | 28.00 | 66.05 | 66.16 | 0.6349 | 82.98 | 2.1206 | -8.70 |
| Mistral-Nemo | 23.43 | 64.63 | 64.60 | 0.6051 | 79.59 | 1.7887 | -5.89 |

## 4 TH-SCORE: A NEW METRIC FOR LLM-AS-A-JUDGE CALIBRATION EVALUATION

While existing calibration metrics such as ECE, Brier Score and NLL offer valuable insights into model reliability, they often overlook practical aspects like high-confidence regions essential for real-world applications in LLM-as-a-Judge scenarios. To address these limitations and better align confidence with accuracy in targeted intervals, we introduce TH-Score, a novel metric designed to improve evaluation for data filtering and quality assessment tasks.

### 4.1 DEFINITION

The TH-Score focuses on two key confidence intervals relevant to practical applications:

- **High-Confidence Data** $(100 - \epsilon, 100)$: These predictions are generally considered highly reliable, and selecting them can significantly enhance the overall dataset quality. $\epsilon$ is a hyperparameter defining the high-confidence threshold.
- **Low-Confidence Data** $(0, \epsilon)$: These predictions are inherently uncertain, and discarding them can effectively reduce noise and enhance overall data quality. $\epsilon$ is also a crucial hyperparameter that determines a threshold for what constitutes low confidence.

This metric quantifies model performance by jointly considering the accuracy of predictions within specified confidence intervals and the coverage of these intervals, facilitating effective data filtering and quality evaluation. The TH-Score is formally defined as:

$$\text{TH-Score} = (e^{(accuracy - 0.5)} - 1) \times percentage,$$

where $e$ denotes the base of the natural logarithm, serving as a scaling hyperparameter; *accuracy* represents the prediction accuracy specifically for samples falling within the target confidence intervals; *percentage* indicates the proportion of total samples that fall within these intervals.

This formulation ensures that the TH-Score increases with both higher accuracy and a larger proportion of high-confidence or low-confidence data, providing a balanced measure of model reliability in practical usage scenarios.

### 4.2 IMPACT OF $\epsilon$ ON TH-SCORE PERFORMANCE

Table 4 presents the TH-Score results for various models under different values of $\epsilon$. The table also includes accuracy rates within specified intervals and the proportion of interval data relative to the total dataset. When $\epsilon = 0.05$, most models, except the most powerful ones, exhibit limited calibration capability. Most models either have minimal data within this interval or demonstrate significantly reduced accuracy, highlighting the stringent calibration demands of such a small $\epsilon$ and underscoring the challenges in achieving reliable confidence alignment at fine-grained thresholds. At $\epsilon = 0.1$,

the value used in our primary experiments, most models align well with this calibration threshold, resulting in strong discriminative power. With the exception of weaker models like Mistral-Nemo, the majority of models have substantial data within this interval, enabling effective comparison of their calibration performance. This observation suggests that an effective approach for selecting $\epsilon$ is to choose a value where most models contribute significant data to the interval.

However, when $\epsilon$ is increased to 0.15, while data coverage improves,the discriminative power diminishes. The advantages of high-performing models, such as DeepSeek-R1-0528, become less pronounced due to the relaxed performance requirements associated with a larger $\epsilon$. Thus, selecting an appropriate $\epsilon$ requires balancing data coverage with discriminative power, avoiding excessively large values that dilute model differentiation.

Table 3: A comparison of calibration metrics. Our proposed TH-Score is designed to evaluate practical reliability in high-confidence regions, addressing the limitations of standard approaches. Notation: % = interval coverage; $\epsilon$ = adjustable threshold (default=0.1); acc = accuracy within $\epsilon$ ranges; $o_i$ = ground truth; $p_i$ = predicted probability.

| Metric | Formula | Key Characteristics |
|---|---|---|
| ECE | $\sum_{i=1}^{M} \frac{n_i}{N} \lvert \text{acc}(i) - \text{conf}(i) \rvert$ | ✗ Fixed-width bins. ✗ Ignores high-confidence regions. |
| ACE | Variant of ECE with adaptive binning | ✗ Computationally expensive. ✗ Lacks focus on confidence intervals. |
| Brier Score | $\frac{1}{N} \sum_{i=1}^{N} (p_i - o_i)^2$ | ✗ Less interpretable. ✗ Fails to isolate miscalibration. |
| MCE | $\max_{i \in \{1,..,M\}} \lvert \text{acc}(i) - \text{conf}(i) \rvert$ | ✗ Sensitive to outliers. ✗ Not reflective of overall calibration. |
| NLL | $-\frac{1}{N} \sum [y_i \log(p_i) + (1 - y_i) \log(1 - p_i)]$ | ✗ Unbounded, hard to compare. ✗ Sensitive to overconfident errors. |
| **TH-Score** | $(e^{(\text{acc}-0.5)} - 1) \times \%$ | ✓ Targets high-confidence regions. ✓ Uses adaptive threshold $\epsilon$. ✓ Balances accuracy and coverage. ✓ Interpretable, bounded score. |

Table 4: Model performance under different $\epsilon$ values in SC setting.

| Model | $\epsilon = 0.05$ | | | $\epsilon = 0.10$ | | | $\epsilon = 0.15$ | | |
|---|---|---|---|---|---|---|---|---|---|
| | Acc ↑ | % ↑ | TH ↑ | Acc ↑ | % ↑ | TH ↑ | Acc ↑ | % ↑ | TH ↑ |
| DeepSeek-R1-0528 | 1.00 | 37.4 | **12.14** | 0.91 | 68.6 | **17.52** | 0.87 | 81.1 | **18.38** |
| Qwen3-235B-A22B | 0.88 | 8.0 | 1.93 | 0.88 | 63.4 | 14.59 | 0.84 | 78.0 | 15.73 |
| GPT-4.1 | 1.00 | 1.1 | 0.37 | 0.83 | 38.0 | 7.55 | 0.69 | 74.3 | 7.88 |
| R1-Distill-Llama | 0.86 | 24.9 | 5.42 | 0.71 | 59.4 | 7.01 | 0.65 | 81.4 | 6.72 |
| GPT-4.1-mini | 0.00 | 0.0 | 0.00 | 0.83 | 13.7 | 2.71 | 0.63 | 68.6 | 4.57 |
| DeepSeek-V3-0324 | 1.00 | 0.6 | 0.19 | 0.79 | 9.4 | 1.57 | 0.65 | 44.3 | 3.63 |
| GPT-4o | 0.00 | 0.0 | 0.00 | 0.71 | 21.4 | 2.46 | 0.54 | 72.3 | 1.38 |
| Llama-3.3-70B | 0.52 | 22.0 | 0.22 | 0.54 | 43.1 | 0.80 | 0.48 | 74.3 | -0.57 |
| GPT-4.1-nano | 0.00 | 0.0 | 0.00 | 0.43 | 2.0 | -0.07 | 0.46 | 6.9 | -0.14 |
| Mistral-Nemo | 0.36 | 12.9 | -0.86 | 0.20 | 88.9 | -11.64 | 0.20 | 94.9 | -12.12 |

## 5 LLM-AS-A-FUSER

As shown in the section on overconfidence in LLM-as-a-judge, while LLM-as-a-judge offers a promising and increasingly practical approach to evaluating diverse model outputs, its calibration issues—such as overconfidence in unreliable judgments—limit its overall reliability. Traditional aggregation methods (e.g., majority voting) compound this problem by ignoring nuanced critiques from individual models and focusing only on final decisions. To address these limitations, we propose **LLM-as-a-Fuser** framework, which redefines the LLM's role from a passive judge to an active

*fuser*. By synthesizing model decisions and their rationales, the fuser enables evidence-aware aggregation, ultimately improving both calibration and robustness. As shown in Figure 4, the fuser ingests decisions and critiques from an ensemble of models, analyzing reasoning. Unlike traditional methods, this approach grounds the final decision in evidence.

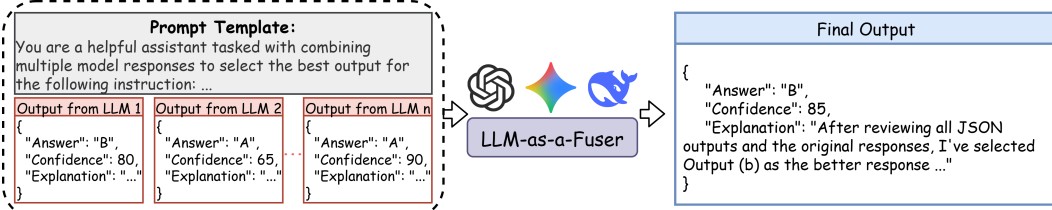

Figure 4: Illustration of the LLM-as-a-Fuser framework, where the fuser model aggregates decisions and their corresponding critiques from an ensemble of models.

## 5.1 BASELINE METHODS

To evaluate the performance of LLM-as-a-Fuser, we compare it against several baseline aggregation methods that combine predictions from multiple models. These methods vary in how they weight or process model predictions and confidences but do not incorporate model critiques, relying solely on final decisions and, where applicable, associated confidence scores. The baseline methods are:

- **Majority Voting**: Selects the most frequent label across models, with equal votes counted. Ties are broken fairly by the highest confidence score.
- **Confidence-Weighted Voting**: Weights votes by model confidence scores, selecting the label with the highest total. Ties use the maximum confidence.
- **Square-Root Confidence-Weighted Voting**: Applies square-root transformation to confidences, summing them to select the label with the highest total.
- **Entropy-Weighted Voting**: Weights model confidences by inverse entropy, ultimately selecting the label with the highest weighted confidence sum.

These baseline methods serve as standard approaches for aggregating model predictions and provide a robust comparison for evaluating the effectiveness of LLM-as-a-Fuser, which leverages model critiques in addition to final decisions. Each method was implemented with careful consideration of model calibration and tie-breaking mechanisms to ensure fair and consistent comparisons.

## 5.2 EXPERIMENTAL RESULTS

Table 5 presents the performance of LLM-as-a-Fuser and baseline on JudgeBench, compared to individual model results under the Self-Confidence (SC) setting in Table 1. LLM-as-a-Fuser with Qwen3-235B-A22B achieves the highest accuracy (86.29%) and best calibration (ECE of 6.42%), outperforming baselines like Entropy Weighted Voting (81.71% Acc, 8.48% ECE) and showing substantial gains over SC models (e.g., +8.86% Acc and -5.36% ECE relative to its SC counterpart at 77.43% Acc, 11.78% ECE). Notably, models like Mistral-Nemo exhibit dramatic improvements (+47.14% Acc, -53.73% ECE), followed by Gemini-2.5-Flash (+38.57% Acc) and GPT-4.1-nano (+30.85% Acc), indicating weaker SC performers benefit significantly from critique integration in the fuser framework. Baseline aggregation methods also surpass individual SC performances; for instance, Entropy Weighted Voting exceeds the top SC model by 4.28% in accuracy and 3.3% in ECE, while other baselines (80% Acc) outperform most SC models. Other fusers vary, with GPT-4o weakest (49.71% Acc, 44.07% ECE). Critique integration drives LLM-as-a-Fuser's accuracy and calibration, and ensemble methods generally yield better results than isolated SC evaluations.

## 5.3 DISAGREEMENT WITH MAJORITY VOTING

We analyzed cases where LLM-as-a-Fuser's decisions diverged from majority voting, as visualized in Figure 5. Qwen3-235B-A22B, the top-performing fuser (Table 5), has the most correct disagreements (34) and few incorrect ones (12), reflecting its effective use of model critiques. In contrast,

Table 5: Performance comparison between baseline aggregation methods and LLM-as-a-Fuser models. Parenthesized values indicate the performance change relative to the SC baseline (Table 1). Green and red colors indicate performance improvement and degradation, respectively.

| Method/Fuser Model | Acc ↑ | ECE ↓ | NLL ↓ | TH ↑ |
|---|---|---|---|---|
| **Entropy W. Voting** | **81.71** | **8.48** | 0.53 | **13.08** |
| Conf. W. Voting | 80.00 | 10.43 | **0.50** | 12.64 |
| Majority Voting | 80.00 | 10.77 | **0.50** | 12.58 |
| Sqrt Conf. W. Voting | 80.00 | 10.43 | **0.50** | 12.64 |
| **LLM-as-a-Fuser** | | | | |
| **Qwen3-235B-A22B** | **86.29** (+8.86) | **6.42** (-5.36) | **0.39** (-0.13) | **17.38** (-0.14) |
| OpenAI-o3-mini | 84.86 (+10.57) | 8.16 (-7.81) | 0.48 (-0.14) | 16.39 (+3.56) |
| GPT-4.1-mini | 83.14 (+27.43) | 10.24 (-22.46) | 0.47 (-0.53) | 16.37 (+13.08) |
| Claude-Sonnet-4 | 81.71 (+17.42) | 9.06 (-8.92) | 0.54 (-0.15) | 12.31 (+2.42) |
| GPT-4.1 | 80.00 (+16.86) | 14.92 (-11.47) | 0.69 (-0.16) | 16.04 (+8.49) |
| DeepSeek-V3-0324 | 78.86 (+29.15) | 12.71 (-23.50) | 0.54 (-0.49) | 11.96 (+9.50) |
| Gemini-2.5-Flash | 78.00 (+38.57) | 15.72 (-14.77) | 0.67 (-0.11) | 13.49 (+10.78) |
| Deepseek-R1-0528 | 68.57 (-8.29) | 21.44 (+9.37) | 1.65 (+1.23) | 10.34 (-4.25) |
| Mistral-Nemo | 67.43 (+47.14) | 20.49 (-53.73) | 0.95 (-2.06) | 13.53 (+25.17) |
| Llama-3.3-70B | 62.86 (+20.86) | 24.38 (-22.99) | 1.39 (-1.36) | 9.80 (+9.00) |
| GPT-4.1-nano | 57.71 (+30.85) | 37.25 (-19.78) | 2.36 (+0.98) | 5.48 (+5.55) |
| GPT-4o | 49.71 (+0.00) | 44.07 (+4.82) | 2.06 (+0.91) | 0.72 (-0.85) |

GPT-4o has the most incorrect disagreements (112) and fewest correct ones (6), indicating poor integration. DeepSeek-V3-0324 shows the fewest total disagreements (30), suggesting conservative decision-making, while Llama-3.3-70B has few correct disagreements (11), aligning with its lower accuracy (62.86%). These results highlight the fuser's ability to leverage critiques for accurate decisions, with Qwen3-235B-A22B's performance underscoring the framework's strength.

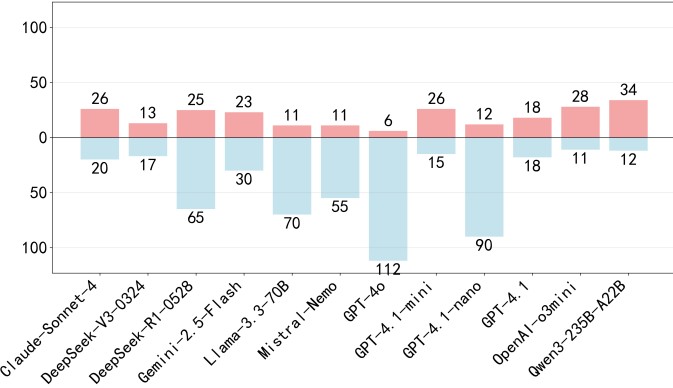

Figure 5: Number of correct (positive bars) and incorrect (negative bars) disagreements between majority voting and the LLM-as-a-Fuser across different models.

## 6  CONCLUSION

This work diagnoses the Overconfidence Phenomenon in LLM-as-a-Judge, where confidence exceeds accuracy, undermining reliability in tasks like data filtering. We introduce TH-Score to quantify calibration in key intervals, offering a practical alternative to metrics like ECE, and propose LLM-as-a-Fuser, an ensemble framework that integrates critiques for enhanced calibration—yielding up to +47.14% accuracy and -53.73% ECE improvements on JudgeBench. These innovations enable confidence-driven, risk-aware evaluations, thereby reducing human oversight while boosting trustworthiness in practical applications. Future directions include investigating the root causes of the overconfidence phenomenon and developing more scalable solutions.

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

# A APPENDICES

## A.1 LLMS USAGE IN OUR WORK

We hereby pledge that, regarding the use of LLMs in this work, we only used them for minor text refinement, apart from their evaluation purposes in the experiments.

## A.2 SELF-CONFIDENCE (SC) SETTING PROMPT

The following is the prompt template used in the Self-Confidence (SC) setting to elicit both the judgment result and confidence score from the LLM. Placeholders such as {{question}}, {{answer_a}}, and {{answer_b}} are replaced with the actual instruction and output pairs during evaluation.

```
SELF-CONFIDENCE (SC) SETTING PROMPT

You are a helpful assistant in evaluating the quality of the
    outputs for a given instruction. Your goal is to select the best
     output for the given instruction and provide a confidence score
     (0-100) for your selection.
Select the Output (a) or Output (b) that is better for the given
    instruction. The two outputs are generated by two different AI
    chatbots respectively.
Evaluate the following outputs, and provide your best guess along
    with a confidence score in the following JSON format:
{
  "selected_output": "Output (a)" or "Output (b)",
  "confidence_score": number,
  "explanation": "Your detailed explanation here"
}
# Instruction:
{{question}}
# Output (a):
{{answer_a}}
# Output (b):
{{answer_b}}
Your response must be in the JSON format as shown above. Do not
    output ANYTHING else. Do not provide the % symbol.
```

## A.3 MULTIPLE-PROMPTING (MP) AND LOGP SETTINGS PROMPT

The following is the prompt template used in the Multiple-Prompting (MP) and LogP settings to elicit the judgment result from the LLM. Placeholders such as {{question}}, {{answer_a}}, and {{answer_b}} are replaced with the actual instruction and output pairs during evaluation.

```
MULTIPLE-PROMPTING (MP) AND LOGP SETTINGS PROMPT

You are a helpful assistant in evaluating the quality of the
    outputs for a given instruction. Your goal is to select the best
     output for the given instruction.
Select the Output (a) or Output (b) that is better for the given
    instruction. The two outputs are generated by two different AI
    chatbots respectively.
Evaluate the following outputs, and provide your best guess in the
    following JSON format:
{
  "selected_output": "Output (a)" or "Output (b)",
  "explanation": "Your detailed explanation here"
```

```
}
# Instruction:
{{question}}
# Output (a):
{{answer_a}}
# Output (b):
{{answer_b}}
Your response must be in the JSON format as shown above. Do not
    output ANYTHING else.
```

## A.4 LLM-AS-A-FUSER PROMPT

The following is the prompt template used in the LLM-as-a-Fuser framework to synthesize judgments from multiple models. Placeholders such as {{question}}, {{answer_a}}, {{answer_b}}, and the Jinja loop for JSON outputs are replaced with actual data during evaluation.

### LLM-AS-A-FUSER PROMPT

```
You are a helpful assistant tasked with combining multiple model
    responses to select the best output for the following
    instruction: Evaluate the quality of multiple outputs for a
    given instruction and select the best one based on specific
    rules.

**Task:**
You will receive:
1. The instruction describing the task.
2. Multiple outputs (e.g., Output (a), Output (b)) generated by
    different models.
3. A list of JSON outputs, each containing:
    - selected_output: The chosen output (e.g., "Output (a)").
    - confidence_score: A score showing the model's confidence (e.g
        ., 85).
    - explanation: Why the model chose that output.

Your goal is to:
- Review the JSON outputs and evaluate the original outputs (Output
     (a), Output (b), etc.) using the evaluation rules.
- Pick the best output or create a new one by combining the best
    parts of multiple outputs.
- Return a JSON response with the selected output, confidence_score
    , and an explanation.

**Input:**
- **Instruction**: {{ question }}
- **Outputs**:
  - Output (a): {{ answer_a }}
  - Output (b): {{ answer_b }}
- **JSON Outputs**:
{% for output in json_outputs %}
  - JSON Output {{ loop.index }}: {{ output }}
{% endfor %}

**Steps:**
1. **Check JSON Outputs**:
    - Look at each selected_output, confidence_score, and
        explanation.
    - Use the explanation to understand why the model picked that
        output.
```

```
     - Note the confidence_score, but focus on explanation quality
        and rule compliance.
  2. **Evaluate Original Outputs**:
     - Judge Output (a), Output (b), etc., against the evaluation
        rules.
     - Use JSON explanations to guide your evaluation.
  3. **Pick or Combine**:
     - Choose the best output if one clearly meets the rules.
     - If no output is perfect, combine the best parts of multiple
        outputs to create a better response.
  4. **Explain Your Choice**:
     - Say why you picked the output or created a new one.
     - Mention the JSON outputs' explanations and scores, noting
        agreements or differences.
     - Show how your choice follows the rules better than others.

  **Output Format:**
  ```json
  {
    "selected_output": "Output (a)" or "Output (b)",
    "confidence_score": number(0-100),
    "explanation": "Why you chose this output or how you combined
        outputs, referencing JSON explanations, confidence scores, and
        evaluation rules."
  }
  ```
```

## A.5 SUPPLEMENTARY TABLES

Table 6: Performance Comparison of Different LLMs under MP Setting

| Model | Acc ↑ | ECE ↓ | ACE ↓ | Brier Score ↓ | MCE ↓ | NLL ↓ | TH Score ↑ |
|---|---|---|---|---|---|---|---|
| DeepSeek-R1-0528 | **85.43** | **7.17** | **6.69** | **0.108** | **70.00** | **0.83** | **17.90** |
| Qwen3-235B-A22B | 78.86 | 13.00 | 12.04 | 0.151 | 70.00 | 0.98 | 16.85 |
| OpenAI-o3-mini | 76.00 | 18.49 | 18.77 | 0.184 | 43.91 | 1.84 | 17.08 |
| R1-Distill-Llama | 71.71 | 15.14 | 15.83 | 0.206 | 60.00 | 1.46 | 7.84 |
| R1-Distill-Qwen | 67.71 | 18.06 | 17.72 | 0.215 | 37.42 | 1.17 | 8.10 |
| Claude-Sonnet-4 | 64.29 | 34.51 | 34.48 | 0.340 | 65.00 | 6.12 | 9.17 |
| GPT-4.1 | 63.14 | 34.91 | 34.96 | 0.346 | 70.00 | 6.04 | 8.27 |
| Gemini-2.5-Flash | 52.57 | 14.43 | 14.77 | 0.220 | 43.33 | 1.44 | 7.40 |
| DeepSeek-V3-0324 | 50.57 | 47.89 | 47.96 | 0.479 | 70.00 | 9.02 | 0.79 |
| GPT-4o | 49.71 | 47.09 | 46.97 | 0.463 | 58.13 | 7.64 | 1.68 |
| GPT-4.1-mini | 56.00 | 42.31 | 42.18 | 0.419 | 65.00 | 7.52 | 4.35 |
| Llama-3.3-70B | 42.86 | 54.31 | 54.28 | 0.537 | 70.00 | 9.49 | -1.82 |
| GPT-4.1-nano | 28.29 | 67.43 | 67.41 | 0.663 | 71.26 | 10.86 | -6.76 |
| Mistral-Nemo | 19.43 | 68.89 | 68.93 | 0.643 | 78.69 | 6.69 | -4.35 |

## A.6 SUPPLEMENTARY FIGURES

Table 7: Complete performance comparison of baseline aggregation methods and LLM-as-a-Fuser models. Values in parentheses represent changes compared to the original Self-Confidence (SC) setting (Table 1). In the LLM-as-a-Fuser section, changes are colored: green indicates improvements (better performance relative to SC baseline, considering metric directions: higher for ↑, lower for ↓), while dark gray indicates deteriorations (worse performance). Neutral changes (e.g., +0.00) are uncolored.

| Method/Fuser Model | Acc ↑ | ECE ↓ | ACE ↓ | MCE ↓ | Brier ↓ | NLL ↓ | TH ↑ |
|---|---|---|---|---|---|---|---|
| **Entropy W. Voting** | **81.71** | **8.48** | **9.4** | 38.51 | **0.15** | 0.53 | **13.08** |
| Conf. W. Voting | 80.00 | 10.43 | 13.0 | 12.98 | 0.16 | **0.50** | 12.64 |
| Majority Voting | 80.00 | 10.77 | 12.9 | 12.89 | 0.16 | **0.50** | 12.58 |
| Sqrt Conf. W. Voting | 80.00 | 10.43 | 13.0 | 12.98 | 0.16 | **0.50** | 12.64 |
| **LLM-as-a-Fuser** | | | | | | | |
| **Qwen3-235B-A22B** | **86.29** (+8.86) | **6.42** (-5.36) | **8.9** (-3.3) | 70.00(+6.50) | **0.12** (-0.04) | **0.39** (-0.13) | **17.38** (-0.14) |
| OpenAI-o3-mini | 84.86 (+10.57) | 8.16 (-7.81) | 9.1 (-8.1) | **21.68**(-18.32) | 0.13 (-0.07) | 0.48 (-0.14) | 16.39 (+3.56) |
| GPT-4.1-mini | 83.14 (+27.43) | 10.24 (-22.46) | 11.8 (-21.0) | 80.00(+35.79) | 0.14 (-0.21) | 0.47 (-0.53) | 16.37 (+13.08) |
| Claude-Sonnet-4 | 81.71 (+17.42) | 9.06 (-8.92) | 10.3 (-7.7) | 65.00(+20.00) | 0.15 (-0.09) | 0.54 (-0.15) | 12.31 (+2.42) |
| GPT-4.1 | 80.00 (+16.86) | 14.92 (-11.47) | 15.6 (-11.2) | 23.96(-31.04) | 0.18 (-0.11) | 0.69 (-0.16) | 16.04 (+8.49) |
| DeepSeek-V3-0324 | 78.86 (+29.15) | 12.71 (-23.50) | 13.5 (-22.9) | 75.00(+24.76) | 0.17 (-0.20) | 0.54 (-0.49) | 11.96 (+9.50) |
| Gemini-2.5-Flash | 78.00 (+38.57) | 15.72 (-14.77) | 16.0 (-14.4) | 33.33(-22.78) | 0.19 (-0.07) | 0.67 (-0.11) | 13.49 (+10.78) |
| Deepseek-R1-0528 | 68.57 (-8.29) | 21.44 (+9.37) | 22.3 (+11.0) | 41.67(-18.33) | 0.24 (+0.11) | 1.65 (+1.23) | 10.34 (-4.25) |
| Mistral-Nemo | 67.43 (+47.14) | 20.49 (-53.73) | 20.5 (-53.7) | 27.79(-52.21) | 0.22 (-0.49) | 0.95 (-2.06) | 13.53 (+25.17) |
| Llama-3.3-70B | 62.86 (+20.86) | 24.38 (-22.99) | 24.8 (-22.0) | 38.85(-24.93) | 0.27 (-0.18) | 1.39 (-1.36) | 9.80 (+9.00) |
| GPT-4.1-nano | 57.71 (+30.85) | 37.25 (-19.78) | 37.4 (-19.7) | 75.00(+2.50) | 0.38 (-0.14) | 2.36 (+0.98) | 5.48 (+5.55) |
| GPT-4o | 49.71 (+0.00) | 44.07 (+4.82) | 44.3 (+5.0) | 48.00(-9.50) | 0.44 (+0.04) | 2.06 (+0.91) | 0.72 (-0.85) |

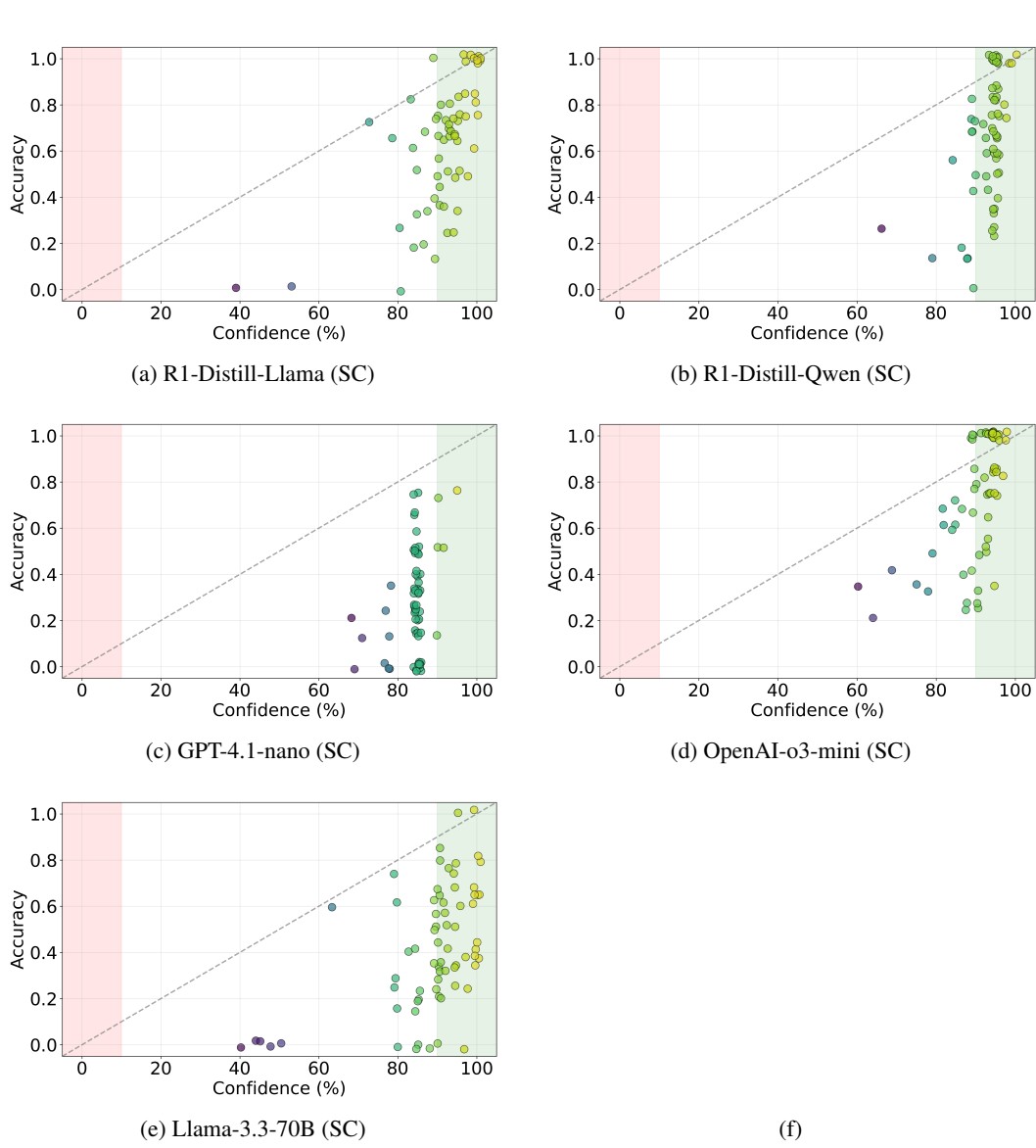

Figure 6: Supplementary figures: SC setting, page 1.

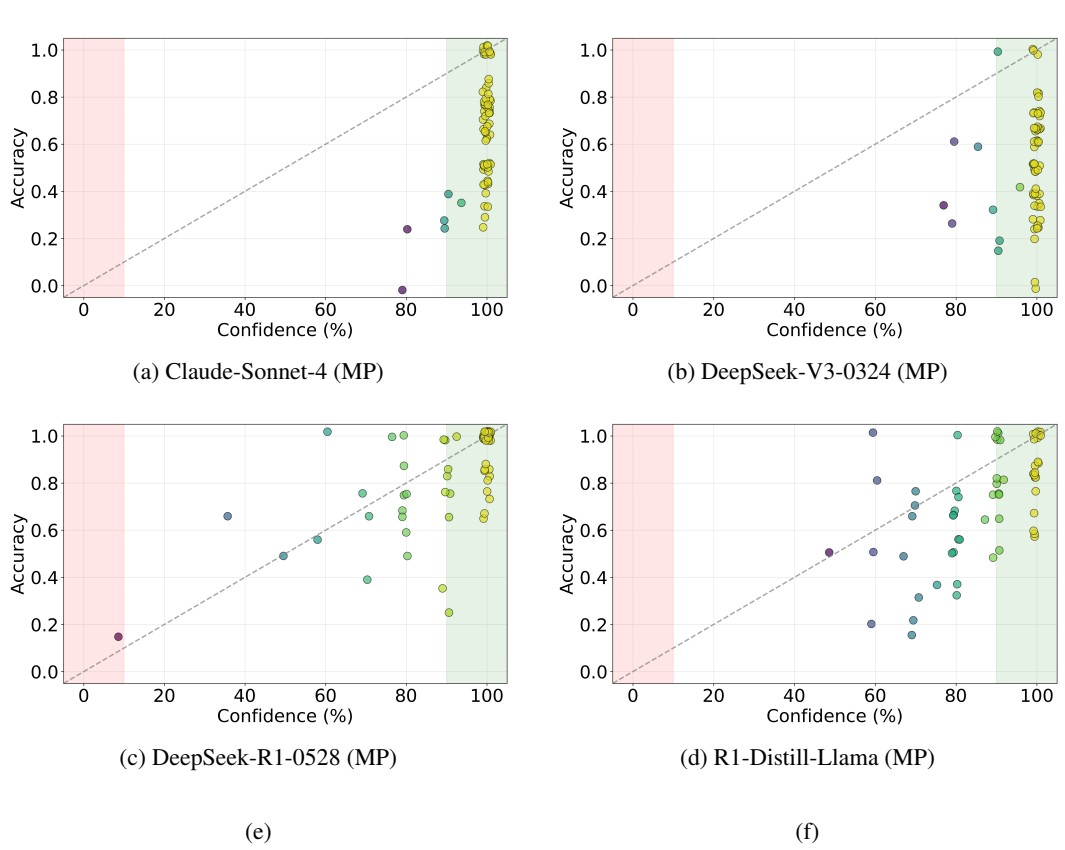

(a) Claude-Sonnet-4 (MP)

(b) DeepSeek-V3-0324 (MP)

(c) DeepSeek-R1-0528 (MP)

(d) R1-Distill-Llama (MP)

(e)

(f)

Figure 7: Supplementary figures: MP setting, page 2.

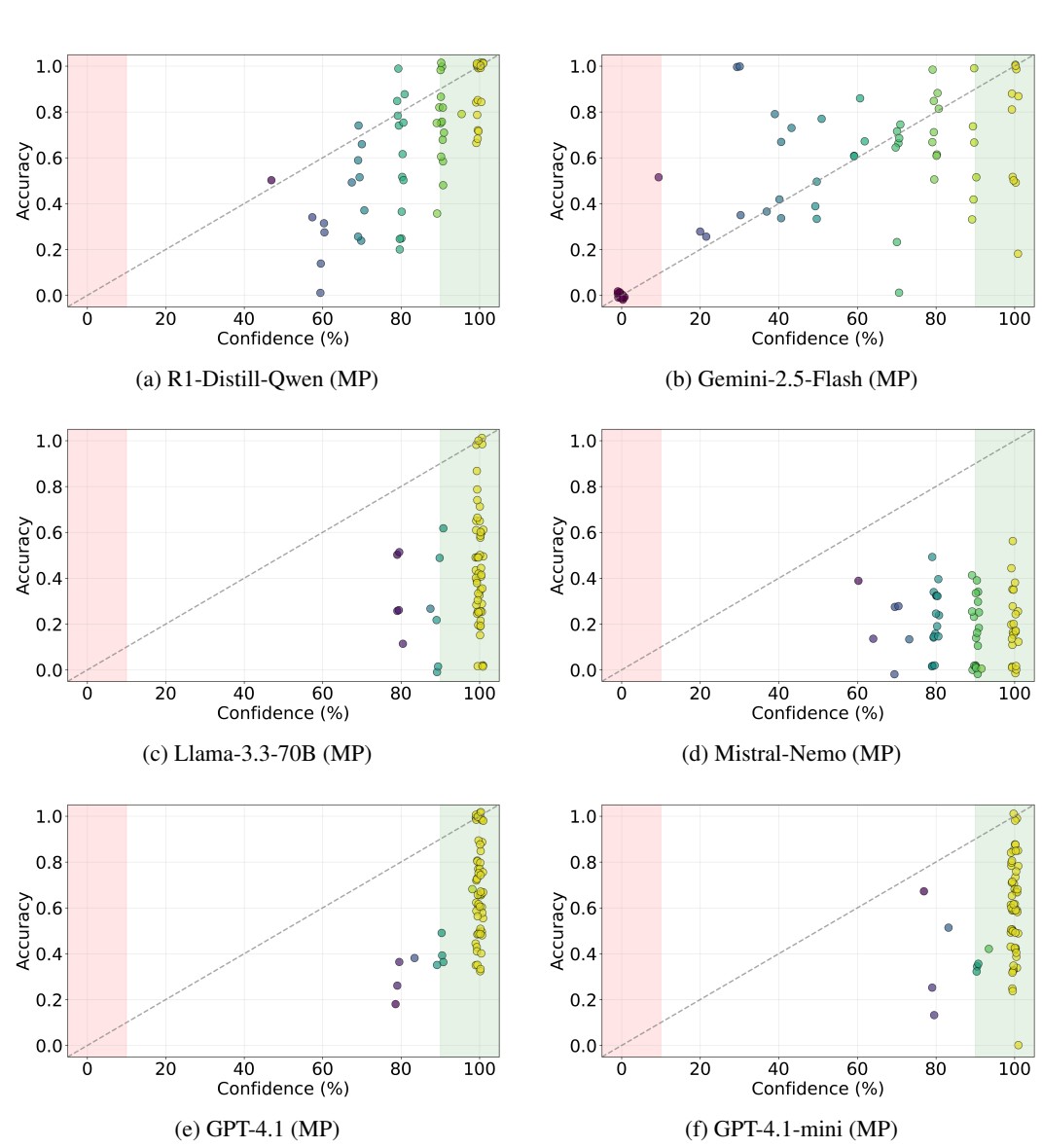

(a) R1-Distill-Qwen (MP)

(b) Gemini-2.5-Flash (MP)

(c) Llama-3.3-70B (MP)

(d) Mistral-Nemo (MP)

(e) GPT-4.1 (MP)

(f) GPT-4.1-mini (MP)

Figure 8: Supplementary figures: MP setting, page 3.

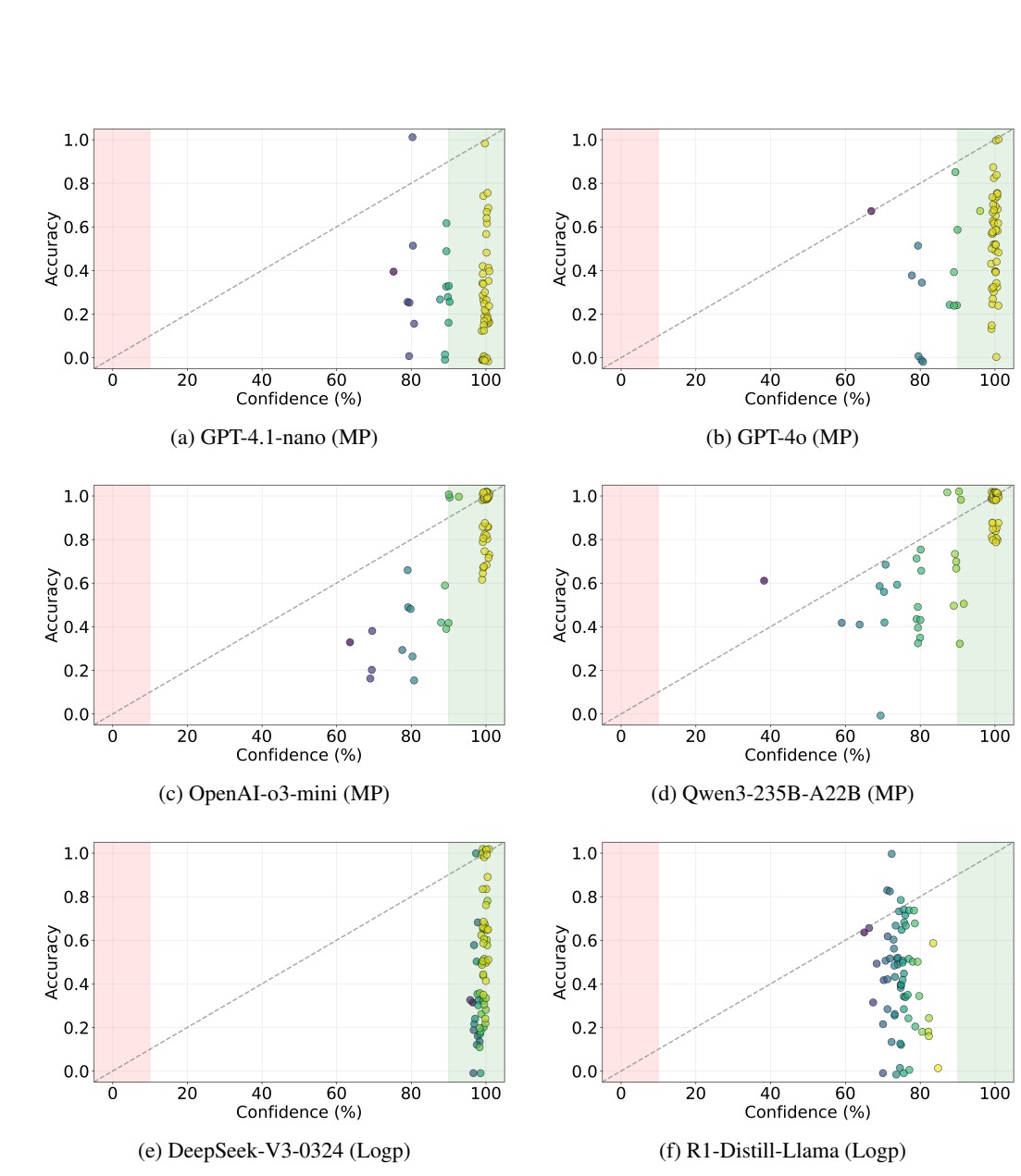

(a) GPT-4.1-nano (MP)

(b) GPT-4o (MP)

(c) OpenAI-o3-mini (MP)

(d) Qwen3-235B-A22B (MP)

(e) DeepSeek-V3-0324 (Logp)

(f) R1-Distill-Llama (Logp)

Figure 9: Supplementary figures: Logp setting, page 4.

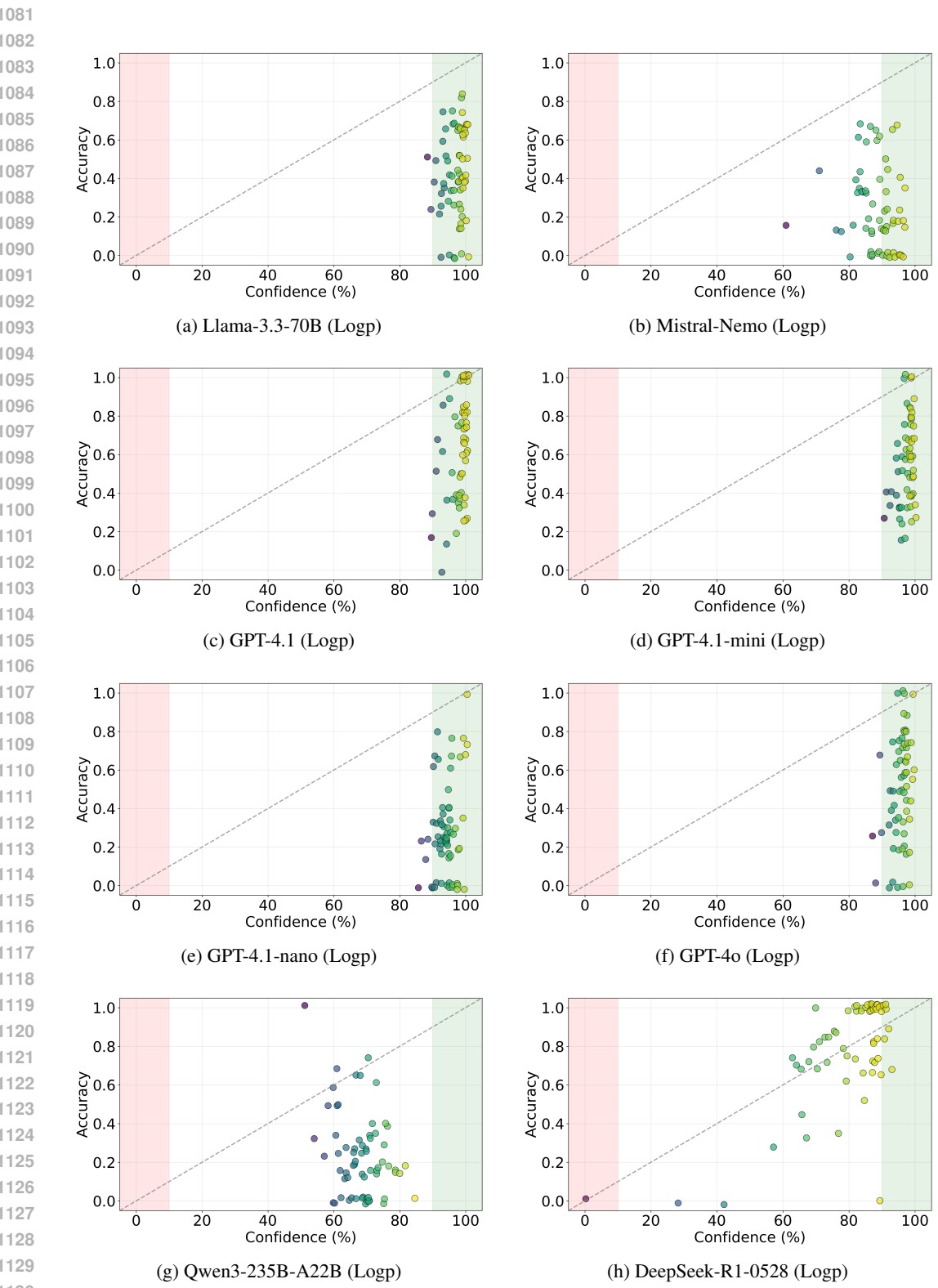

Figure 10: Supplementary figures: Logp setting, pages 5.

