# OpenReview forum: "Overconfidence in LLM-as-a-Judge: Diagnosis and Confidence-Driven Solution"
_ICLR.cc/2026/Conference — Submitted to ICLR 2026_

### Official Review · Reviewer_FxZX · 2025-10-18

**Soundness:** 1
**Presentation:** 3
**Contribution:** 2
**Rating:** 2
**Confidence:** 4

**Summary:**

This paper investigates the problem of overconfidence in the LLM-as-a-Judge paradigm. The authors diagnose this "Overconfidence Phenomenon" and propose a new metric, the TH-Score, to measure the alignment between a model's confidence and its actual accuracy. Furthermore, they introduce LLM-as-a-Fuser, an ensemble framework designed to integrate judgments from multiple models to improve calibration and overall performance.

**Strengths:**

The paper addresses a critical and timely issue. Within the LLM-as-a-Judge framework, large language models—particularly the most powerful ones—have a tendency to produce judgments that are both overconfident and incorrect. Therefore, improving the calibration between confidence and accuracy is a crucial step toward building more reliable and trustworthy evaluation systems, and this work makes a valuable contribution in that direction.

**Weaknesses:**

1. **Lack of Experimental Validation for TH-Score**: A notable weakness is the lack of direct experimental validation for the proposed TH-Score. While the authors claim that the metric better captures confidence-accuracy alignment compared to standard metrics like ECE and NLL, the paper **does not provide experiments** that empirically substantiate this claim. This omission makes it difficult to assess the metric's effectiveness and advantages over existing, widely-used calibration measures.

2. **Applicability of TH-Score to Different Confidence Methods**: In Section 4.1, the definition of TH-Score relies on confidence intervals near **0 and 1**, controlled by a parameter $\epsilon$. This formulation seems well-suited for the Self-Confidence (SC) method, where confidence scores span the full [0, 1] range. However, its applicability to the other described confidence methods is unclear. For instance, the confidence derived from Log Probability (LogP) has a theoretical minimum of 0.5. The paper does not present results for TH-Score under the Multiple-Prompting (MP) or LogP settings, which raises questions about the metric's generalizability beyond the SC method.

**Questions:**

1. Table 4 presents a sensitivity analysis of the TH-Score to different values of $\epsilon$. Could the authors clarify how these results should guide the practical selection of this hyperparameter? The paper notes that an effective approach is to "choose a value where most models contribute significant data to the interval," but a more principled method or further discussion on how to determine the optimal setting for $\epsilon$ would strengthen the work.

2. On L205, the paper mentions "R1-Distill-Qwen" and "R1-Distill-Llama". Could the authors please provide more specific details about these models (e.g., base model, parameter count) for clarity and to aid reproducibility?

---

### Official Review · Reviewer_q3Q7 · 2025-10-30

**Soundness:** 2
**Presentation:** 3
**Contribution:** 2
**Rating:** 4
**Confidence:** 3

**Summary:**

The paper try to addresses the overconfidence phenomenon in LLMs when used as automated judges, where the models’ predicted confidence significantly overstates their actual correctness, undermining the reliability of practical evaluations. The authors introduce TH-Score, a novel metric that quantifies confidence-accuracy alignment by focusing on critical high- and low-confidence intervals, and propose LLM-as-a-Fuser, an ensemble framework that uses a dedicated "fuser" LLM to synthesize judgments and critiques from multiple models to enhance calibration. Extensive experiments on 14 cutting-edge LLMs using the JudgeBench benchmark show that LLM-as-a-Fuser substantially improves calibration and accuracy—with the top-performing fuser (Qwen3-235B-A22B) achieving 86.29% accuracy and 6.42% ECE, outperforming baseline aggregation methods—and TH-Score effectively detects the overconfidence phenomenon, ultimately promoting more trustworthy LLM-as-a-Judge systems.

**Strengths:**

* It clearly defines and systematically analyzes the "Overconfidence Phenomenon" in the scenario of Large Language Models as judges (LLM-as-a-Judge) — where the model’s predicted confidence significantly exceeds its actual correctness.

* The experimental design is relatively rigorous, covering 14 mainstream LLMs (including open-source and closed-source models) and adopting three confidence calculation methods

**Weaknesses:**

* TH-Score Lacks Generalization and Mechanistic Validation：Only evaluated on JudgeBench, but LLM-as-a-Judge benchmarks (MTBench, FairEval, LLMBar) vary in task types (pairwise vs. single-sample) and evaluation criteria (subjective style vs. objective logic).

* Unjustified hyperparameter ε: Selects ε=0.1 as optimal but provides no analysis of how ε performs across different task subdomains of JudgeBench (e.g., math vs. coding). For example, math tasks may require stricter high-confidence thresholds (smaller ε) than coding ?

* LLM-as-a-Fuser’s Mechanism Is Opaque and Unvalidated, No ablation for "reason fusion"，does not compare Fuser to a variant that only uses model decisions. methods lack innovation.

* Unproven Practical Value (No Cost-Benefit Validation):it provides no quantitative cost-benefit analysis, How much additional compute does Fuser require (vs. single-model judges or baselines)?

**Questions:**

NO ETHICS STATEMENT and REPRODUCIBILITY STATEMENT
Please see the weaknesses.

---

### Official Review · Reviewer_4GHb · 2025-10-30

**Soundness:** 2
**Presentation:** 2
**Contribution:** 2
**Rating:** 2
**Confidence:** 3

**Summary:**

The paper studies calibration for LLM-as-a-Judge settings and reports the overconfidence phenomenon. It introduces (1) TH-Score, a confidence-interval–focused metric intended to capture alignment between confidence and accuracy in low/high-confidence regions, and (2) LLM-as-a-Fuser, an ensemble scheme that aggregates multiple model judgments and critiques via a ``fuser'' LLM. On JudgeBench, the authors claim sizable calibration and accuracy gains over individual judges and standard voting.

**Strengths:**

1. Overconfidence is an important and underexplored issue in LLM-as-a-Judge research. Addressing it has clear practical and scientific significance.
2. The paper successfully motivates why calibration matters for judge models, especially in scenarios where high-confidence predictions may replace human evaluation.

**Weaknesses:**

1. Lack of detail and limited novelty in the LLM-as-a-Fuser method. The main algorithmic contribution is not clearly described. From what is presented, the fusion process seems to rely on standard majority or weighted voting with an additional critique prompt, which limits its methodological originality. A more thorough description, including architecture, prompt examples, and ablation studies (with/without critiques, number of judges, different model combinations), would strengthen this part.

2. Limited empirical validation of TH-Score. The paper introduces TH-Score as a key metric but does not convincingly demonstrate its validity or advantages over established calibration measures such as ECE, Brier Score, or risk–coverage curves.

- 2.1 The authors should analyze the correlation between TH-Score and other metrics.
- 2.2 They should also discuss situations where TH-Score offers unique insight or better discriminative ability.
- 2.3 Sensitivity to the ϵ parameter should be more systematically studied.

3. Writing and presentation issues.

- 3.1 All figures and tables should be placed at the top of their pages for readability.
- 3.2 The description of TH-Score (currently in Section 4) should be moved earlier in the paper to improve narrative flow.
- 3.3 Some variable values appear inconsistent: according to Line 200, the values in Lines 319–323 should be 5 and 10, not the current numbers.

**Questions:**

Please see the Weaknesses section

---

### Official Review · Reviewer_Jkx3 · 2025-10-31

**Soundness:** 1
**Presentation:** 2
**Contribution:** 2
**Rating:** 2
**Confidence:** 4

**Summary:**

This paper addresses a critical limitation of the "LLM-as-a-Judge" paradigm: the Overconfidence Phenomenon, where state-of-the-art LLMs (e.g., GPT-4o, Mistral-Nemo) exhibit predicted confidence far exceeding actual accuracy (e.g., GPT-4o’s ECE of 39.25 under Self-Confidence setting), undermining reliable risk-aware evaluation.

To tackle this, the work presents two core innovations:

1.TH-Score：A novel metric quantifying confidence-accuracy alignment in practically critical intervals (high-confidence: 100−ε to 100; low-confidence: 0 to ε, ε=0.1 by default). It balances interval accuracy and sample coverage, addressing the limitations of traditional metrics (e.g., ECE ignoring key intervals, Brier Score’s poor interpretability).

2. LLM-as-a-Fuser: An ensemble framework that uses a dedicated "fuser" LLM to synthesize decisions
and their rationales from multiple source LLMs—unlike baseline aggregation methods (majority voting, confidence-weighted voting) that only use final judgments.

Experiments on the JudgeBench benchmark (350 pairs across knowledge, reasoning, math, coding) validate effectiveness: LLM-as-a-Fuser with Qwen3-235B-A22B achieves 86.29% accuracy and 6.42% ECE (outperforming Entropy-Weighted Voting’s 81.71% accuracy, 8.48% ECE), with weaker models (e.g., Mistral-Nemo) gaining +47.14% accuracy and -53.73% ECE.

This work advances confidence-driven LLM-as-a-Judge systems, providing a tool (TH-Score) for overconfidence diagnosis and a framework (LLM-as-a-Fuser) for adaptive, trustworthy evaluation.

**Strengths:**

Critical Practical Issue Targeting: Systematically identifies and characterizes the "Overconfidence Phenomenon" in LLM-as-a-Judge—a long-overlooked flaw where LLMs (e.g., GPT-4o, Mistral-Nemo) exhibit confidence far exceeding actual accuracy. This addresses a key gap in existing accuracy-centric research, as uncalibrated confidence undermines risk-aware applications (e.g., auto-approving high-confidence judgments).

Innovative Practical Metric (TH-Score): Proposes TH-Score to quantify confidence-accuracy alignment, focusing on task-critical intervals (high: 100−ε to 100; low: 0 to ε). Unlike traditional metrics (ECE, Brier Score) that ignore key intervals or lack interpretability, it balances interval accuracy and sample coverage, directly supporting real-world decision-making.

Novel Evidence-Aware Ensemble Framework: Develops LLM-as-a-Fuser, an ensemble method that synthesizes both decisions and rationales from multiple LLMs via a dedicated "fuser" model. This outperforms baseline aggregation (majority voting, confidence-weighted voting) that only uses final judgments, with Qwen3-235B-A22B fuser achieving 86.29% accuracy and 6.42% ECE—surpassing entropy-weighted voting (81.71% accuracy, 8.48% ECE).

Practical Deployment Alignment: Advocates a paradigm shift to confidence-driven LLM-as-a-Judge systems, enabling adaptive pipelines (auto-approving high-confidence outputs, flagging low-confidence for review). Both TH-Score (overconfidence diagnostic) and LLM-as-a-Fuser (reliable evaluation) are designed for usability, fitting existing workflows.

**Weaknesses:**

Inadequate Technical Precision and Notation Clarity:
The paper lacks rigor in technical details and notation. Critical assertions like "while Bayesian methods are computationally infeasible" are made without citing supporting literature, weakening their credibility. The acronym "TH" in TH-Score is never defined, creating ambiguity about the metric’s conceptual origin. The TH-Score is misleading: "accuracy" specifically refers to the accuracy of
targeted confidence intervals
(high/low-confidence regions) and "percentage" denotes the sample coverage of these intervals, yet neither is distinguished with specialized notation (e.g., acc_interval vs. acc_global). Experimental notation is also inconsistent—Table 4 uses "Acc" to represent interval-specific accuracy without clarifying this in the caption, and Table 5 omits explanations for baseline abbreviations, forcing readers to infer their definitions.

Superficial Method Design and Unjustified Component Inclusion:
The proposed methods are conceptually simple with insufficient elaboration, and key components lack motivational grounding. LLM-as-a-Fuser’s integration of "rationales" is abrupt and poorly aligned with the paper’s core focus on "confidence": the rationale module is not justified as a natural extension of confidence-driven calibration, nor are baselines augmented with rationales for fair comparison. This raises questions about whether the framework’s performance gains stem from confidence modeling or the uncalibrated addition of rationales—suggesting potential unfair advantage over baselines.

Insufficient Experimental Validation and Missing Ablation Studies:
The experimental design fails to validate the framework’s modular dependencies. No ablation studies are conducted for LLM-as-a-Fuser to isolate the impact of critical components: there is no comparison of results using different confidence metrics (e.g., LogP, MP instead of SC), nor validation of performance when confidence is excluded entirely. This absence prevents verification of whether confidence—the link between TH- Score and LLM-as-a-Fuser—adds tangible value, undermining the claim of a cohesive solution.

Lack of Cohesion Between Core Components:
The paper reads as two disjointed works rather than an integrated study. TH-Score and LLM-as-a-Fuser are only superficially connected via "confidence," yet the necessity of confidence in this relationship is unproven. Structurally, the paper segregates their methods and experiments without cross-referencing insights. Experimentally, TH-Score is evaluated across three confidence settings (SC, MP, LogP), but LLM-as-a-Fuser exclusively uses SC, ignoring the broader findings from the metric’s validation and breaking methodological consistency.

Opaque Experimental Setup:
Critical experimental details are omitted, compromising result reproducibility and interpretation. For Table 5, the paper does not specify which source models the baselines (e.g., Majority Voting, Entropy-Weighted Voting) aggregate, nor clarify whether listed models (Qwen3-235B-A22B, GPT-4o) function as "fuser" models—and if so, which source LLMs they synthesize. These omissions obscure how experimental design impacts outcomes, as aggregation performance is highly sensitive to both source model diversity and fuser capability.

Shallow Result Analysis and Avoidance of Anomalies:
Result interpretation is limited to numerical comparison, lacking critical depth and engagement with anomalies. For instance, Table 1 shows Qwen3-235B-A22B (TH-Score=17.52) outperforming Mistral-Nemo (TH-Score=-11.64), but the paper only attributes this to "better calibration in high-capability models" without analyzing
why
model scale or architecture drives this gap. Anomalous results are ignored: Table 5 shows Qwen3-235B-A22B’s TH-Score decreasing with Fuser, Deepseek-R1-0528 and GPT-4o performing worse as fusers, and Figure 5 depicts excessive erroneous disagreements in GPT-4o—yet none of these are explained, nor linked to the framework’s limitations (e.g., fuser’s poor rationale integration). This evasion undermines the analysis’s academic rigor.

Unsubstantiated Claims of Academic Contribution:
The paper fails to validate its purported advantages over prior work. It cites Yang et al. (2024) noting that multi-model aggregation requires substantial resources but provides no analysis of whether LLM-as-a-Fuser reduces computational or data costs. Without comparative resource benchmarks, the framework’s practical or academic noveltyin addressing resource constraints remains unproven.

Neglected Limitations and Generalization Boundaries:
No systematic discussion of limitations or generalization is provided. Experiments are restricted to a single benchmark (JudgeBench), with no exploration of how the methods perform in high-stakes closed domains (e.g., medical diagnosis). The paper also does not address whether TH-Score’s ε threshold or LLM-as-a-Fuser’s effectiveness requires domain-specific tuning—leaving the reliability and applicability of conclusions in real-world scenarios unsubstantiated.

**Questions:**

1.For the critical assertion that "Bayesian methods are computationally infeasible" in the context of LLM calibration, why does the paper not cite any supporting literature to validate this claim, and how might this omission affect the credibility of the paper’s choice to avoid Bayesian approaches?

2.The acronym "TH" in TH-Score is never defined in the paper. What does "TH" stand for, and why was this conceptual origin of the metric not clarified to avoid ambiguity for readers?

3.In the TH-Score, "accuracy" specifically refers to interval-specific accuracy (for high/low-confidence regions) and "percentage" refers to interval sample coverage. Why were these terms not distinguished with specialized notation (e.g., acc_interval vs. acc_global)?

4.Table 4 uses "Acc" to represent interval-specific accuracy (not global accuracy), but this is not clarified in the table caption. Why was this key distinction omitted, and how might it lead to misinterpretation of the TH-Score’s performance across different ε values?

5.Table 5 includes baselines like "Entropy W. Voting" and "Conf. W. Voting" but does not explain these abbreviations. Why were these baseline definitions missing, and what steps could have been taken to improve the table’s readability?

6.LLM-as-a-Fuser integrates "rationales" of source models, but the paper does not justify how this component aligns with the core focus on "confidence-driven calibration." What motivated the addition of rationales, and why were baselines not augmented with rationales to ensure a fair comparison of the framework’s performance gains?

7.The paper claims LLM-as-a-Fuser’s advantages stem from confidence-aware aggregation, yet no ablation studies test performance when using different confidence metrics (e.g., LogP, MP instead of SC) or when confidence is excluded entirely. Why were these ablation studies omitted, and how can the paper prove that confidence (not just rationales) is critical to the framework’s success?

8.TH-Score is evaluated across three confidence settings (SC, MP, LogP), but LLM-as-a-Fuser exclusively uses SC. Why was this methodological inconsistency introduced, and how does it impact the coherence between the metric’s validation and the framework’s experimental design?

9.Table 5 compares LLM-as-a-Fuser with baseline aggregation methods (e.g., Majority Voting), but it does not specify which source models these baselines aggregate, nor clarify if listed models (e.g., Qwen3-235B-A22B) act as "fusers" and which source LLMs they synthesize. Why were these critical experimental details omitted?

10.Table 1 shows Qwen3-235B-A22B (TH-Score=17.52) outperforming Mistral-Nemo (TH-Score=-11.64), but the paper only attributes this to "better calibration in high-capability models." What specific aspects of model scale (e.g., parameter size) or architecture (e.g., training data, reasoning modules) drive this calibration gap, and why were these factors not analyzed?

11.In Table 5, Qwen3-235B-A22B’s TH-Score decreases when used as a fuser, and Deepseek-R1- 0528/GPT-4o perform worse as fusers. Why do these anomalies occur, and how do they relate to potential limitations of LLM-as-a-Fuser (e.g., poor rationale integration for certain models)?

12.Figure 5 shows GPT-4o has excessive erroneous disagreements when deviating from majority voting. The paper claims this reflects "poor integration ability," but it does not link this to the framework’s design. How do GPT-4o’s erroneous disagreements specifically reveal flaws in LLM-as-a-Fuser’s rationale synthesis or confidence weighting?

13.The paper cites Yang et al. (2024) noting that multi-model aggregation requires substantial resources, but it provides no analysis of whether LLM-as-a-Fuser reduces computational/data costs compared to prior methods like Collaborative Calibration. Why was this comparative resource benchmark omitted?

14.Experiments are restricted to the JudgeBench benchmark, with no testing in high-stakes closed domains (e.g., medical diagnosis). Why were these domains not included, and how can the paper assess LLM-as-a-Fuser’s generalization to scenarios where calibration errors have severe consequences?

15.The paper does not discuss whether TH-Score’s ε threshold or LLM-as-a-Fuser’s effectiveness requires domain-specific tuning. How might ε need to be adjusted for tasks like medical text evaluation (vs. general translation), and why was this tuning analysis not conducted to define the method’s generalization boundaries?

---

### Meta-Review · Area_Chair_1UkT · 2025-12-24

**Summary:**

While all reviewers found the problem the authors focus on relevant, they raised many points that remained unaddressed as no response was given.

The reviewers where concerned about the lack of technical rigor and found the writing was vague to the degree where the main algorithm was not clearly described and experimental details where missing, potentially hindering reproducibility. The experimental evaluation was deemed not sufficient, particularly concerning was the missing ablations (i.e. for the fusion). Further, the authors could have evaluated on more benchmarks. Also the limited novelty was concerning.

**Reviewer Concerns:**

No concerns where addressed as no rebuttal was provided.

**Reviewer Scores:**

As no reply by the authors was provided, the scores would not have changed.

---

### Decision · Program_Chairs · 2026-01-26

Reject